# New Insights into Mucosa-Associated Microbiota in Paired Tumor and Non-Tumor Adjacent Mucosal Tissues in Colorectal Cancer Patients

**DOI:** 10.3390/cancers16234008

**Published:** 2024-11-29

**Authors:** Adriana González, Asier Fullaondo, David Navarro, Javier Rodríguez, Cristina Tirnauca, Adrian Odriozola

**Affiliations:** 1Department of Genetics, Physical Anthropology and Animal Physiology, University of the Basque Country UPV/EHU, 48940 Bilbao, Spain; adriana.gonzalez@ehu.eus (A.G.); asier.fullaondo@ehu.eus (A.F.); 2Danagen-Bioted S.L., 08915 Barcelona, Spain; david@danagen.es; 3Department of Oncology, Clínica Universidad de Navarra, 31008 Pamplona, Spain; jrodriguez@unav.es; 4Department of Mathematics, Statistics and Computer Science, University of Cantabria, 39005 Santander, Spain; cristina.tirnauca@unican.es

**Keywords:** colorectal cancer (CRC), gut microbiota, tissue samples, personalized medicine, 16S rRNA gene, MinION sequencing, NCBI taxonomic classification

## Abstract

Colorectal cancer (CRC) is a major burden of disease worldwide. Increasing scientific evidence highlights the role of the gut microbiota in the initiation, development and treatment of CRC. Currently, the analysis of CRC-associated gut microbiota has several limitations that hinder its implementation in precision medicine, including selection of sample type, sequencing platform and taxonomic classification. This article aims to address these constraints to provide data on CRC-associated microbiota and facilitate the implementation of its analysis in personalized medicine. To this end, mucosa-associated microbiota from paired tumor and non-tumor adjacent tissue samples from 65 CRC patients was analyzed through V3–V4 region of 16S rRNA gene amplification, MinION sequencing and NCBI taxonomic classification. Results consistent with available evidence have been obtained. Moreover, to our knowledge, this is the first study that identifies the possible association between a higher relative abundance of *Streptococcus periodonticum* and a lower relative abundance of *Corynebacterium* with CRC.

## 1. Introduction

Colorectal cancer (CRC) is one of the most common cancers and a major global burden of disease. In 2022, more than 1.9 million new cases and more than 900,000 deaths were estimated worldwide due to this disease. Despite advances in prevention, diagnosis and treatment, by 2045, the incidence and mortality of CRC are projected to increase by 70.5% and 83.4%, respectively [1].

Increasing scientific evidence has shown that human gut microbiota plays a critical role in CRC pathogenesis [2,3]. The gut microbiota, the community of microorganisms symbiotically inhabiting our gut, can be modulated by several lifestyle factors, among which diet stands out [4]. It is estimated that a 70% reduction in CRC incidence could be achieved by following a healthy and balanced diet [5,6]. Scientific evidence suggests that the key link between CRC and diet may lie in the gut microbiota [7,8].

Gut microbiota composition imbalance, known as dysbiosis, contributes to the start and progression of CRC. Previous works have identified CRC key pathogens, such as *Fusobacterium nucleatum* [9,10], *Streptococcus gallolyticus* [11], enterotoxigenic *Bacteroides fragilis* [12], *Peptostreptococcus anaerobius* [13] and *Clostridioides difficile* [14]. These pathogens may contribute to colorectal carcinogenesis directly by damaging DNA and stimulating colonocyte proliferation or indirectly by promoting a favorable environment for CRC development [15,16]. Otherwise, some bacteria in the gut microbiota exert protective effects against CRC. The main mechanisms by which the gut microbiota plays a key role in colorectal carcinogenesis include intestinal permeability regulation, inflammation and immune response modulation, biofilm formation, genotoxin production, virulence factors, oxidative stress and metabolite production [17].

There are two hypotheses regarding the role of the gut microbiota in colorectal carcinogenesis. The “alpha-bug” hypothesis suggests that certain pro-oncogenic microorganisms can displace cancer-protective bacteria and colonize the tumor persistently, creating an environment favorable for tumorigenesis [18]. On the other hand, the driver-passenger model suggests that “driver” bacteria that initiate CRC are then replaced by “passenger” bacteria with growth advantages in the tumor microenvironment (TME) that may exert promoting or inhibiting effects on the tumor progression [19].

Due to scientific evidence supporting its influence on CRC pathogenesis, gut microbiota is becoming increasingly important in personalized medicine, which relies on patients’ genetic and molecular characteristics to adapt the therapeutic strategy [20].

Consequently, we hypothesized that microbiota composition would differ between paired tumor and non-tumor adjacent tissues of CRC patients and between tumor locations and that analyzing these possible differences would help better understand how the tumor microbiota influences the pathogenesis of CRC. Specifically, (1) we compared paired tumor and non-tumor adjacent tissues from 65 Spanish CRC patients, and (2) we examined differences in microbial composition concerning the tumor location.

However, several challenges in analyzing the CRC-associated microbiota limit its implementation in personalized medicine, such as sample type, the sequencing platform, and the database for bacterial taxonomic classification [21].

Regarding sample type, most published studies examining microbiota composition in CRC have analyzed fecal samples. Although easy to obtain, fecal samples contain microbiota from different intestinal locations and do not adequately reflect microbiota interactions within the intestinal epithelium and TME as direct tumor tissue samples do. However, despite the importance of local CRC tissue-associated microbiota, there is a lack of solid direct information on tumor tissue samples [21].

Regarding sequencing platforms, microbiota analysis is limited in personalized medicine due to the lack of access to expensive and time-consuming conventional technology [22]. In this context, the sequencing platform MinION from Oxford Nanopore Technologies (ONT) implies a revolution exhibiting translational potential in clinical practice due to its portability, low cost and real-time sequencing [23,24].

Finally, the taxonomic assignment of 16S ribosomal RNA (rRNA) gene reads is commonly carried out by SILVA, RDP or Greengenes. While those tools map well on NCBI, the contrary is problematic because NCBI comprises more taxa and sequences and is continuously updated and curated. As a result, NCBI shows higher effectiveness in classifying 16S rRNA gene reads than the other available tools [25].

Overall, this study aims to overcome the main challenges in CRC-associated microbiota analysis, including sample type, the sequencing platform, and the database for bacterial taxonomic classification, to enhance our understanding of CRC-associated gut microbiota alteration and facilitate the implementation of their analysis in personalized medicine.

To this end, we amplified V3–V4 regions of the 16rRNA bacterial gene by the sequencing platform MinION and performed the taxonomic assignment using the NCBI Taxonomy Database.

## 2. Materials and Methods

### 2.1. Ethics Statement and Sample Collection

One hundred and thirty paired tumor and non-tumor adjacent tissue samples were obtained from 65 CRC patients of the Biobank of the University of Navarra, Spain. Samples and data from patients included in the study were provided by the Biobank of the University of Navarra and were processed following Standard Operating Procedures approved by the Ethical and Scientific Committees of Clinica Universitaria de Navarra (CUN) for the research (REINFORCE_0011-1411-2020-000102). All individuals gave written informed consent.

Samples from tumor and non-tumor adjacent mucosal tissues were obtained from each CRC patient by biopsy forceps during endoscopy and tumor removal surgery in the CUN. The pathologist selected, if possible, a fragment of tumor tissue and a fragment of non-tumor adjacent mucosal tissues. A Biobank technician, working in sterile conditions and with the material on dry ice, cut the selected tissue into small 2–3 mm square fragments placed in a cryotube for immediate freezing in dry ice. All were registered and stored at −80 °C at the Biobank until DNA extraction. 

The general information (age and gender) and clinical data (tumor location, tumor differentiation and tumor stage) of samples are shown in Appendix A. For the 65 CRC patients, the origin of the paired tissue samples was diverse: the colon for 27, the rectum for 28, the sigmoid colon for 9 and the cecum for 1.

### 2.2. DNA Extraction and Quantitation

DNA was extracted using the Danagene Microbiome Tissue DNA kit (Danagen-Bioted S.L., Barcelona, Spain). DNA quantitation was carried out by fluorometry (Qubit 2.0, Life Technologies, Carlsbad, CA, USA, Thermo Fisher Scientific, Waltham, MA, USA) using HS dsDNA Assay (ThermoFisher Scientific, Waltham, MA, USA) and by spectrophotometry (NanoDrop 2000c, Thermo Fisher Scientific, Waltham, MA, USA). Negative DNA extraction controls were included.

### 2.3. PCR Amplification

PCR amplification of the 450 base pair (bp) V3–V4 region of the 16S rRNA gene was conducted using the Molzym Mastermix 16S complete DNA-free kit (Molzym, Bremen, Germany). Amplification was performed using an Applied Biosystems Veriti^TM^ Thermal Cycler (Thermo Fischer Scientific, Waltham, MA, USA) according to the manufacturer’s protocol. Negative PCR amplification controls were included.

### 2.4. Library Preparation and Amplicon Sequencing

PCR products were purified using magnetic beads of the Clean NGS reagent (CleanNA, Waddinxveen, The Netherlands). Then, each mixture was quantified by fluorometry (Qubit 2.0, Life Technologies, Carlsbad, CA, USA, ThermoFisher Scientific, Waltham, MA, USA) using HS dsDNA Assay (ThermoFisher Scientific, Waltham, MA, USA) to calculate the DNA input for library preparation.

Multiplex MinION sequencing was carried out using 16S rRNA gene amplicons (SQK-NBD114.96; Oxford Nanopore Technologies, Oxford, UK). In addition, 130 ng DNA per sample was used for amplicon library preparation. As different clinical samples were combined into pooled libraries to homogenize conditions and obtain comparable results, barcoded adapters were used. During this process, end-repair procedures and adapter ligation were carried out. MinION^TM^ sequencing was performed using a MinION nanopore DNA sequencer (MIN-101B) and Flow Cell R10 (FLO-MIN114) according to the manufacturer’s instructions (Oxford Nanopore Technologies, Oxford, UK).

### 2.5. Data Acquisition and Sequencing Data Analysis

MINKNOW UI software version 23.04.3 (Oxford Nanopore Technologies, Oxford, UK) was used for data acquisition and base-calling converting sequence reads (i.e., FAST5 data) into FASTQ files by Guppy version 6.5.7 pipeline (Oxford Nanopore Technologies, Oxford, UK).

The Barcoding workflow in the Metrichor Ltd. analysis platform EPI2ME (Oxford Nanopore Technologies, Oxford, UK) was used for taxonomic classification. For that purpose, FASTQ files were uploaded to the EPI2ME desktop agent 16S workflow (Oxford Nanopore Technologies, Oxford, UK), where real-time classification was carried out using the NCBI 16S rRNA gene blast database. Reads were filtered for Q-score ≥ 9. The MinION was run for up to 48 h. Results were processed by in-house software to avoid infra representation of each taxonomical ID and convert reads in relative abundance according to the estimated 16S gene copy number (GCN) for each taxon based on the rrnDB database version 5.9 [26].

### 2.6. Bioinformatics Analysis

For α-diversity analysis, community richness was calculated using the number of observed Operational Taxonomic Units (OTUs) and the Chao1 index, whereas diversity and evenness were analyzed by calculating the Shannon–Weaver index [27] and the Simpson index [28] using Python 3.11. Data visualization for α-diversity results was performed with box plots using Python 3.11. For β-diversity analysis, two metrics for OTUs relative abundance were generated: Bray–Curtis dissimilarity and Jaccard index and the correspondent matrices. The β-diversity results were plotted using PAST 4.13 in a Principal coordinates analysis (PCoA). Bioinformatic analysis was performed to establish specific qualitative and quantitative microbiota compositions between groups using GraphPad 8.0. Data visualization for differential abundance analysis was performed with stacking maps and violin plots using SRPlot [29].

### 2.7. Statistical Analysis

A two-tailed paired *t*-test was performed to compare α-diversity mean differences between tumor and non-tumor tissue groups for the number of observed OTUs, Chao1, Shannon and Simpson indexes. The Mann–Whitney U test was performed to compare α-diversity mean differences between colon, rectum and sigmoid colon tumor tissue locations. A *p*-value below 0.05 was considered to be statistically significant.

To evaluate differences in β-diversity, we used Analysis of Similarities (ANOSIM) and Permutational Multivariate Analysis of Variance (PERMANOVA) tests using two metrics, Bray–Curtis dissimilarity and Jaccard index, in PAST 4.13.

Multiple *t*-tests were used to evaluate OTUs relative abundance differences between tumor and non-tumor tissue groups in GraphPad Prism 8.0. The Benjamini, Krieger and Yekutieli method for controlling the False Discovery Rate (FDR) was used to consider multiple comparisons. FDR-adjusted *p*-values below 0.01 were considered statistically significant.

## 3. Results

### 3.1. Sequence Analysis

We analyzed the microbiota composition of paired tumor and non-tumor adjacent tissue samples from 65 CRC patients. 27,305,189 raw reads were analyzed through 16S rRNA gene sequencing with a mean length of 637.37 ± 13.2 bp, and an average quality score of 10 ± 0.36. Two paired tissue samples were excluded because no reads were obtained for the non-tumor sample. A total of 25,193,582 reads were assigned to the remaining 128 samples. After quality filtering, 18,878,209 high-quality reads from the 128 samples were obtained. The average reads per sample for the tumor and non-tumor tissues were 155,274 ± 188,803 and 136,836 ± 160,153, respectively (*p* = 0.421) (Figure 1). Overall, 3879 different OTUs were identified at a 97% similarity threshold.

### 3.2. α- and β-Diversity

#### 3.2.1. α-Diversity of Microbiota in Tumor Compared to Non-Tumor Adjacent Tissue Samples of CRC Patients

We observed higher community richness (number of OTUs, Chao1 index) and diversity and evenness (Shannon and Simpson indexes) in the non-tumoral compared to the tumoral group.

Three thousand eight hundred and seventy-nine OTUs were identified, with 127 OTUs shared among non-tumor and tumor tissue samples (Figure 1). The average number of OTUs (mean value ± standard error) in the non-tumor and tumor tissue was 503 ± 180 and 488 ± 178, respectively (*p* = 0.499).

A comparison between non-tumor and tumor tissue only reported statistically significant differences for Shannon Index (5.73 ± 0.69 and 5.41 ± 1.03, respectively; *p* = 0.014), while no statistically significant differences were reported for Chao1 (277 ± 127 and 268 ± 123, respectively; *p* = 0.565) or Simpson indexes (0.93 ± 0.09 and 0.95 ± 0.03, respectively; *p* = 0.069) (Figure 2).

#### 3.2.2. β-Diversity of Microbiota in Tumor Compared to Non-Tumor Adjacent Tissue Samples of CRC Patients

The present study analyzed β-diversity between tumor and non-tumor tissue using PCoA of two metrics (Bray–Curtis and Jaccard) and ANOSIM and PERMANOVA analysis (Figure 3). According to Bray–Curtis dissimilarity, the mucosal microbiota composition differed between tumoral and non-tumoral groups. No statistically significant differences were obtained for the Jaccard index. Six samples were classified as outliers by Bray–Curtis dissimilarity and consequently removed for α-diversity, taxa differential abundance and different tumor location analyses.

### 3.3. Specific Microbiota Compositional Differences Between Tumor and Non-Tumor Adjacent Tissue Samples

Significant relative abundance variations were observed in the microbiota of tumor and non-tumor tissue samples at different taxonomic bacterial levels (Table 1 and Table 2).

The phylum Bacillota, Bacteroidota, Pseudomonadota (former Proteobacteria) and Actinomycetota, common members of the human gut microbiota, formed more than 90% of bacterial phyla in tumor and non-tumor tissue samples (Figure 4a). The relative abundances at the phylum level in the tumor and non-tumor tissue samples were compared. We observed a significant difference in four detected phyla between sample groups. The relative abundance of Actinomycetota, Bacteroidota and Pseudomonadota was significantly higher in the non-tumor than in tumor samples. In contrast, Fusobacteriota was significantly higher in tumor samples (Figure 4b).

Seven taxa showed statistically significant differences between tumor and non-tumor tissue samples at the class level (Table 1 and Table 2). The relative abundance of Bacilli and Fusobacteriia in the tumor tissues was significantly higher than in the non-tumor tissues. The relative abundance was significantly lower for the tumor tissues of Actinomycetes, Alphaproteobacteria, Bacteroidia, Betaproteobacteria, and Clostridia than the non-tumor tissues.

Four taxa showed statistically significant differences between tumor and non-tumor tissue samples at the order level (Table 1 and Table 2). Fusobacteriales and Lactobacillales were significantly higher in the tumor compared to the non-tumor tissues. In contrast, Bacteroidales, Eubacteriales, and Mycobacteriales were significantly decreased in tumors compared to the non-tumor tissues.

The microbiota composition also differed at the family level, with seven significantly different families between tumor and non-tumor tissues (Table 1 and Table 2). The relative abundance of Fusobacteriaceae, Leptotrichiaceae and Streptococcaceae was significantly higher in the tumoral group than in the non-tumoral group. The relative abundance of Bacteroidaceae, Corynebacteriaceae, Lachnospiraceae and Propionibacteriaceae was significantly lower in the tumoral than in the non-tumoral group.

*Bacteroides*, *Streptococcus*, *Fusobacterium*, *Prevotella* and *Blautia* were the fifth most abundant genus in tumor and non-tumor tissue samples. Interestingly, among the 20 most abundant genera in tumor and non-tumor tissues, 17 were common, while *Leptotrichia*, *Granulicatella* and *Campylobacter* were also found in tumor tissues and *Lachnospira*, *Clostridium* and *Dorea* in non-tumor tissues (Figure 5a).

The relative abundance of three genera was significantly higher in tumor tissue samples (Table 1 and Table 2): *Fusobacterium*, *Leptotrichia* and *Streptococcus*. In contrast, two genera, *Bacteroides* and *Corynebacterium*, were significantly reduced (Figure 5b).

In tumor tissue samples, species assigned to *F. nucleatum*, *Fusobacterium polymorphum* and *Streptococcus periodonticum* had significantly higher relative abundances than non-tumor samples. Two strains, *F. polymorphum* ATCC 10953 and *Fusobacterium animalis* ATCC 51191, were significantly more abundant in tumor samples.

### 3.4. Tissue-Associated Microbiota Differences Related to Tumor Location in CRC Patients

We observed relevant differences in the composition of the microbiota according to location: colon, rectum and sigmoid colon. After removing the outliers and the only sample from the cecum, the analysis included 25 tumor samples from the colon, 24 from the rectum and 9 from the sigmoid colon. Evaluation of α-diversity between CRC tumor tissue samples from the colon, rectum and sigmoid colon revealed no significant differences (Figure 6).

The analysis of the ANOVA test for β-diversity showed significant differences between colon and rectum tumors, colon and sigmoid colon tumors, and rectum and sigmoid colon tumors for Jaccard Index but not for Bray–Curtis dissimilarity. No statistically significant differences were obtained for the PERMANOVA test (Table 3).

At the taxonomic level, there were significant differences between colon and rectum tumors, colon and sigmoid colon tumors, and rectum and sigmoid colon tumors.

Our results indicated that tumoral microbiota from the colon compared to the rectum was characterized by a preponderance of *Prevotella*, *Roseburia*, *Granulicatella*, *Leyella stercorea*, *Agathobacter rectalis*, *Phocaeicola plebeius* and *Granulicatella elegans* (Appendix A). In contrast, colon tumors compared to rectum showed a decrease in *Alistipes*, *F. animalis*, *F. nucleatum*, *F. polymorphum*, *Fusobacterium vincentii* and *S. periodonticum* (Appendix A).

Concerning tumors from the colon compared to the sigmoid colon, there was an increase in *Roseburia*, *Prevotella*, *Granulicatella*, *L. stercorea* and *G. elegans* (Appendix A). In contrast, colon tumors compared to the sigmoid colon showed a decrease in *Peptoniphilus*, *Staphylococcus*, *Streptococcus* and *Fusobacterium*. At the species level, colon tumors showed a decrease in *F. nucleatum*, *F. polymorphum*, *F. vincentii*, *Peptoniphilus lacrimalis*, *Phocaeicola coprocola*, *Porphyromonas endodontalis*, *S. periodonticum* and *Waltera intestinalis* compared to sigmoid colon tumors (Appendix A).

For tumors from the rectum compared to the sigmoid colon, there was an increase in *Fusobacterium* and *F. animalis* (Appendix A). In contrast, rectum tumors compared to sigmoid colon showed a decrease in *Peptoniphilus*, *Staphylococcus*, *Streptococcus*, *P. lacrimalis*, *P. coprocola*, *S. periodonticum* and *W. intestinalis* (Appendix A).

## 4. Discussion

The development of CRC is associated with genetic and environmental factors, among which diet stands out [5,6]. The available scientific evidence suggests that the link between diet and CRC lies in the gut microbiota [7,8]. A growing body of scientific research has recently shown that gut microbiota can directly affect colorectal tumorigenesis [30].

In the present research, we analyzed the microbiota composition of the paired tumor and non-tumor adjacent tissue samples of the large intestine of CRC patients by a large amplicon, including the V3–V4 regions of 16S rRNA gene with MinION sequencing platform and by NCBI taxonomic classification. This innovative methodology allowed for the finding of significant variations in the relative abundance of bacteria at different taxonomic levels between tumor and non-tumor tissues. These findings are largely consistent with previous studies, and simultaneously, the methodology favors revealing new key taxa even at lower taxonomic levels, such as species and strains [21,31].

The relative abundance of phylum Fusobacteriota (*p* < 10^−6^), genera *Fusobacterium* (*p* < 10^−6^), *Leptotrichia* (*p =* 2 × 10^−6^), and *Streptococcus* (*p* < 10^−6^) was significantly higher in tumor samples. On the other hand, the relative abundance of phyla Actinomycetota, Bacteroidota, and Pseudomonadota (*p* < 10^−6^), and genus *Bacteroides* (*p* < 10^−6^) and *Corynebacterium* (*p* = 8.6 × 10^−4^) was significantly lower in tumor samples. Moreover, although it is necessary to approach these results cautiously, this study showed a higher abundance in tumor samples of *F. nucleatum* (*p* < 10^−6^), *F. polymorphum (p* = 5.3 × 10^−3^), *S. periodonticum* (*p* < 10^−6^) species and *F nucleatum* subsp. *polymorphum* ATCC 10953 and *F. animalis* ATCC 51191 strains (*p* < 10^−6^) compared to non-tumor tissue samples. Despite the scientific interest of those results, those at species and especially at strain levels should be taken cautiously and replicated in future studies, considering the limitations inherent to the available amplicon length and the sequence databases resolution. To the best of our knowledge, the present study is the first to identify the possible increased relative abundance of *S. periodonticus* and decreased *Corynebacterium* association with CRC.

On the other hand, bacteria of the phylum Fusobacteriota were more predominant among the tumor tissue than non-tumor tissue samples, as previously reported [32,33,34]. Fusobacteriota is an understudied phylum of bacteria, including Fusobacteriaceae and Leptotrichiaceae families, both enriched in tumor compared to the non-tumor tissues, consistent with past studies [35]. *Fusobacterium* is a genus included in the first family and *Leptotrichia* in the second one, which was also increased in tumor tissue [36,37]. Notably, several studies have pointed out that the combination of *Fusobacterium* and *Leptotrichia* may contribute to the progression of CRC [38]. 

The molecular mechanisms through which *Fusobacterium* and especially genus members, such as *F. nucleatum* and *F. animalis*, influence CRC are increasingly well described, whereas the ones for *Leptotrichia* are still poorly known [39]. *Leptotrichia* invasive infections have been reported in patients receiving high-dose chemotherapy [40] and in oral disease in immunocompromised patients [41]. One of the most studied species from the genera is *Leptotrichia trevisanii*, an anaerotolerant, anaerobic, opportunistic gram-negative pathogen. Its ability to produce mainly lactic acid from glucose fermentation distinguishes it from closely related *Fusobacterium* species [41]. *Leptotrichia trevisanii* has been described as a causal agent of severe sepsis in immunocompetent patients, particularly in patients with hematological malignancies receiving chemotherapy [42].

According to previous studies, *F. nucleatum* was significantly enriched in the tumor compared to the non-tumor tissues [32,43,44]. *F. nucleatum* is an anaerobe gram-negative opportunistic pathogen ubiquitous in the human oral microbiota [45]. The oral cavity contributes to CRC tumor seeding [46]. Independently of clinical, pathological and molecular features, the amount of *F. nucleatum* in CRC tissue has been positively associated with mortality [47]. *F. nucleatum* has been related to genetic and epigenetic lesions in CRC tissues. *F. nucleatum* is an invasive and proinflammatory microorganism capable of stimulating CRC cell proliferation through different mechanisms: proliferation and metabolism promotion, immune microenvironment reprogramming, proinflammatory microenvironment creation, anticancer immune responses inhibition and metastasis and chemoresistance promotion in CRC [48]. These carcinogenic effects could be mediated by components such as lipopolysaccharides and adhesins like FadA and Fap2 [35]. Hence, *F. nucleatum* is considered a diagnosis biomarker, a prognostic predictor and a promising target in CRC treatment [35,36].

The *F. nucleatum* subspecies genetic and phenotypic heterogeneity has prompted research into differential genetic attributes among species contributing to CRC initiation and progression [49]. In this regard, currently, there is a controversy about the taxonomic classification of *F.nucleatum*. Previous genetic analyses have reported that it comprises five subspecies usually found in oral microbiota: *F. nucleatum* subsp. *nucleatum*, subsp. *polymorphum*, subsp. *vincentii*, subsp. *fusiforme*, and subsp. *animalis* [50]. However, the taxonomic positions of members have been under discussion. In 2010, it was proposed that *F. nucleatum* subsp. *vincentii* and subsp. *fusiforme* could be classified into one subspecies [51]. In 2017, it was proposed to elevate the four *F. nucleatum* subspecies (subps. *nucleatum*, subps. *polymorphum*, subps. *vincentii*, and subps. *animalis*) to species-level status with a wide diversity of significant strains [52].

In the present study, *F. polymorphum* showed higher relative abundance in tumor than non-tumor tissue samples. Recently, *F. polymorphum* has been isolated from CRC patients in their tumor tissue and oral cavity [53] and pre-CRC patients’ saliva [54]. Therefore, to our knowledge, *F. polymorphum* has not been reported to be significantly enriched in CRC tissue compared to non-tumor tissue, and hence, more studies are needed to confirm these findings. The information scarcity could be because it was only in 2017 that *F. polymorphum* was considered a species rather than a subspecies of *F. nucleatum* [52]. 

On the other hand, *F. animalis* is considered the most common and pathogenic species in CRC [55,56]; it also presented a higher relative abundance in tumor tissue than in non-tumor tissue in our study. The presence of *F. animalis* has been previously associated with right-sided tumor location and advanced tumor stages (stages II and III), with higher CRC-specific mortality, and with specific mutations in somatic genes in CRC. In contrast, *Fusobacterium vincentii* or *F. nucleatum* was not and may be driven by a stage shift and chemoresistance [55]. The interaction of *F. animalis* ATCC 51191 with human intestinal epithelial and tumor cells has been verified [57]. Recently, *F. animalis* has been suggested as a primary target for mechanistic and therapeutic studies in CRC [49].

We observed that *Streptococcus* and *S. periodonticus* relative abundance was higher in tumor than in non-tumor tissue samples. To the best of the author’s knowledge, this is the first time that the association of *S. periodonticus* in CRC has been found. *S. periodonticus* is a gram-positive coccus within the genus *Streptococcus* that has been recently isolated from human subgingival dental plaque of periodontitis lesion [58]. *S. periodonticus* has been associated with a pediatric case of bacterial meningitis after cranial surgery [59]. Interestingly, *Streptococcus* spp. internalization into epithelial cells could be facilitated by the effect of *Fusobacterium*, which promotes coaggregation and facilitates internalization processes in these normally non-invasive bacteria [60,61]. Interestingly, a bacterium of the genus *Streptococcus*, *S. gallolyticus*, is a proinflammatory species remarkably associated with CRC. This bacterium can express collagen-binding proteins such as Pil 1, allowing it to colonize tissues and induce the secretion of proinflammatory mediators that can promote CRC [11].

We observed that the relative abundance of phylum Actinomycetota, Bacteroidota and Pseudomonadota was significantly higher in non-tumor than in tumor tissues. Past studies have reached similar results [33,62,63,64]. However, some studies have reported enrichment of the phylum Pseudomonadota in tumor tissue [65,66]. Although a Pseudomonadota increase is generally considered an intestinal dysbiosis marker and gut commensals with pathogenic potential [67], bacteria of the same taxonomic group may exert different effects depending on functional characteristics, interactions and environment [63].

We observed that *Bacteroides* was enriched in non-tumor compared to tumor tissues, which is in coherence with several previous studies [68,69,70]. However, Gao et al. observed that *Bacteroides* was highly enriched in tumor tissues [63]. *Bacteroides* is thought to have both positive and negative impacts on host health via their colitogenic or probiotic effects [71].

*Corynebacterium* genus showed higher relative abundance in non-tumor compared to tumor tissues. To the best of our knowledge, there is no previous information related to this genus in CRC tissue. Regarding cancer, the evidence is scarce and contradictory; one study reported that oxidative tryptamine dimers from *Corynebacterium durum* exert anticancer properties [72], whereas another study reported its contribution to induced colitis [73].

Differences in the microbial profiles observed between tumor and non-tumor tissues could reflect the role of certain microbiota members in the initiation and development of CRC or changes associated with the TME. Two main hypotheses have been proposed to explain these interactions. 

The “alpha-bug” hypothesis proposed by Sears and Pardoll suggests that detrimental members of the microbiota, such as enterotoxigenic *Bacteroides fragilis* (ETBF), *S. gallolyticus*, superoxide-producing *Enterococcus faecalis* and *Escherichia coli*, may act as cancer initiators by directly inducing alterations in colonic epithelial cells and remodeling the colonic microbial community to promote these alterations and disrupt immune responses [18].

In contrast, the bacterial driver-passenger model proposed by Tjalsma et al. suggests that certain commensal bacteria (bacterial drivers), such as *Enterococcus faecalis*, can cause epithelial DNA damage, leading to CRC initiation. Next, the process of tumorigenesis induces modifications in the microenvironment that benefit relatively poor colonizing bacteria (bacterial passengers) [19]. Bacterial passengers have been proposed to include opportunistic pathogens that feed in the tumor (*Fusobacterium* or *Streptococcus* spp.), intestinal commensal or probiotic bacteria (Coriobacteriaceae family), and bacteria that have a competitive advantage in the TME [19].

The microbial α-diversity refers to within-sample diversity. Although no significant differences between groups were obtained for observed OTUs, Chao1 and Simpson indices, the Shannon index was significantly reduced in tumor than in non-tumor tissue samples. From our study, we can conclude that tumor samples present a lower microbial diversity. Previous studies revealed similar results [33,65,74], no statistically significant differences [68,75,76,77], and even reported that microbial α-diversity was significantly higher in the tumor samples [78]. Therefore, there is no consensus between α-diversity metrics behavior in tumor vs. non-tumor tissues.

The microbial β-diversity refers to between-sample diversity. In this study, Bray-Curtis dissimilarity based on OTUs revealed that the microbiota composition of the tumor tissues could be differentiated from non-tumor tissues. In contrast, no statistically significant differences were observed with the Jaccard index. Some previous studies have revealed similar results [33,64,66,69], while others reported no statistically significant differences [74,76,79,80]. These discrepancies could be caused by the Bray–Curtis dissimilarity being based on the relative abundances of the OTUs, while the Jaccard index is based on their presence/absence. 

On the other hand, we identified associations between gut microbiota composition and tumor location. Although there were no significant differences in α and β-diversity analysis, we identified an enrichment of *L. stercorea* and *G. elegans* in colon tumors compared to the rectum and the sigmoid colon. Likewise, the sigmoid colon showed a higher relative abundance of *Peptoniphilus*, *Staphylococcus*, *Streptococcus*, *P. lacrimalis*, *P. coprocola*, *S. periodonticum* and *W. intestinalis* than the colon and the rectum. The rectal tumors were enriched in *F. animalis* compared to the colon and the sigmoid colon. Our results align with previous evidence, supporting that characterization of the microbiota composition at different sites in the large intestine can contribute to a better description of the molecular subtypes of CRC [81,82].

As mentioned, CRC-associated microbiota research has key limitations, such as the sample type selection, the sequencing platform used and the taxonomy classification employed.

Regarding sample type, the present study analyzed the tissue CRC-associated microbiota from 130 paired tumor and non-tumor adjacent tissue samples, in contrast to traditional microbiota analysis from fecal samples [21]. Fecal samples are commonly used because they are easy to collect, non-invasive and repeatable, which makes them potential tools for CRC screening and early detection. In addition, they are less likely to be contaminated by eukaryotic DNA [83,84]. Fecal microbiota can provide valuable information, but it primarily reflects the composition of the intestinal lumen microbiota, and it does not capture the same perturbations and interactions with the colonic mucosa critical to CRC pathogenesis [85,86].

In contrast, as analyzed in the present study, mucosa-associated microbiota interacts more closely with colonocytes and host local immunity, possibly playing a greater role in CRC initiation and development than luminal microbiota. These interactions can lead to gene expression modifications and inflammation, which may influence colorectal tumorigenesis [87]. Therefore, tissue samples are ideal for investigating the pathogenesis of CRC. Moreover, tissue samples are considered more accurate for bacterial detection, show higher microbial diversity than fecal samples, and are more suitable for identifying potentially minor alterations in the microbiota of precancerous colorectal lesions [84]. Despite the potential importance of tissue samples, studies examining the differential abundance of associated microbiota in tumor and non-tumor tissue in CRC patients are scarce [21].

However, the analysis of tissue samples showed additional practical challenges. While in fecal samples, our group usually analyzed the whole 16S rRNA gene (approximately 1500 bp) from fecal samples [88], after several attempts in tissue samples, the maximum amplicon size reached was approximately 450 bp corresponding V3–V4 region of the 16SrRNA gene. Similar limitations have been observed when investigating mucosal microbiota with shotgun metagenomics [89]. We hypothesized that studying the complete 16SrRNA gene amplicon in tissue samples is unfeasible due to human DNA’s high presence and the target gene’s fragmentation. Nevertheless, the 450 bp reached in the current article comprises significantly larger amplicons than those usually obtained by sequence platforms different from MinION (200–300 bp), improving the potential taxonomic profiling [90]. The larger read size combined with NCBI taxonomic classification improves the taxonomic profiling.

On the other hand, due to operative limitations, the present study did not include a CRC-free control group because of the difficulty in obtaining colon samples from healthy people. However, using paired tumor and non-tumor samples avoids the potential confounding factors when analyzing CRC and healthy controls, such as age, sex, body mass index, diet and other characteristics [21,82].

Otherwise, tissue sampling is more invasive than fecal, limiting the number of samples collected from the tumor and non-tumor tissues in the present study. It would be advisable to analyze more samples from each patient for future research to examine the possible heterogeneity of the microbiota composition associated with CRC tumor tissue [76]. In addition, future longitudinal studies looking at changes in the microbiota over time would provide a better understanding of whether specific bacteria play a role in the initiation and progression of CRC or whether they are responding to TME. Finally, considering the pros and cons of analyzing fecal or tissue samples, combining both seems valuable for future research.

On the other hand, selecting a taxonomic classification database is essential for analyzing CRC-associated microbiota. The most used taxonomic classifications for 16S rRNA gene studies vary in size (taxa amount) and resolution capacity (classification level). The current scientific evidence shows that the taxonomic classification at the genus and species level is essential to decoding the role of microbiota composition in CRC. In the present article, we have selected the NCBI database because it contains the largest number of nodes and the highest resolution. Thus, NCBI allows a more thorough classification, going down to the species rank and below and offering several intermediate ranks [25].

In contrast, SILVA and RDP are limited to classifying the genus level as the lowest rank. Although Greengenes goes down to species, the comparative analysis carried out by Balvočiūtė & Huson showed that SILVA, RDP and Greengenes map well with NCBI but not vice versa. Hence, they recommended using the NCBI taxonomy as a common framework for 16S rRNA gene studies [25]. In addition, the NCBI taxonomy is manually curated, covering more than 150 sources, and updated daily [91].

Nevertheless, taxonomy selection up to date is generally determined by the pipeline used, and most studies that analyze tumor microbiota composition in CRC patients use SILVA, RDP or Greengenes. Therefore, it dramatically limits the resolution and amplitude of the previous results and reanalyzing them using the NCBI database could improve the available scientific evidence regarding CRC-associated microbiota.

Regarding sequence platform selection in CRC research, access to sequencing technology is an important limitation for implementing CRC-associated gut microbiota analysis in precision medicine [22]. MinION from ONT is a revolutionary sequencing platform because of its small size and portability, and it allows real-time sequencing of various samples on demand at a competitive cost [23,24]. In the present article, MinION is used for its potential to facilitate the implementation of microbiota analysis in precision medicine strategies for CRC. Despite its advantages, to the best of our knowledge, the previous evidence using the MinION sequencer for microbiota analysis of paired tumor and non-tumor tissues in CRC patients is limited to a single article [22]. In this article, the authors did not carry out 16S rRNA gene amplicon sequencing with MinION but used MinION sequencing using genomic DNA to analyze the microbiota. They conclude that long-read sequences generated using MinION allow differentiation between bacterial strains and plasmids, and as a cost-effective and rapid sequencing tool, it has the potential for use in clinical settings [22].

## 5. Conclusions

In this study, we detected an enrichment in genera such as *Fusobacterium*, *Leptotrichia* and *Streptococcus* in tumor compared to non-tumor tissue samples. In addition, species such as *F. nucleatum*, *F. polymorphum*, *S. periodonticus* and strains such as *F. polymorphum* ATCC 10953 and *F. animalis* ATCC 51191 were also enriched in tumor tissue. On the other hand, genera such as *Bacteroides* and *Corynebacterium* were enriched in non-tumor tissues. In addition, differences in microbiota composition were observed between tumor locations (colon, rectum and sigmoid colon).

The present study faces the main technical challenges in CRC-associated microbiota regarding sample type, sequence platform, and taxonomic database. It comprises the analysis of 130 paired tumor and non-tumor adjacent tissue samples in different locations by larger amplicon sizes, including the V3–V4 region of 16S rRNA gene analyzed by MinION suitable sequence platform, and using NCBI database, that increases the amplitude and resolution in sequence taxonomy classification. These methodological advances agree with earlier findings of microbiota composition differences between tumor and non-tumor tissues. They also reveal the possible involvement of specific taxa, such as *S. periodonticus* and *Corynebacterium*, in CRC biology for the first time.

Therefore, the results obtained in this study facilitate the implementation of individual gut microbiota analysis in personalized medicine. This approach allows the development of therapeutic strategies for CRC that consider this essential component of TME [20].

## Figures and Tables

**Figure 1 cancers-16-04008-f001:**
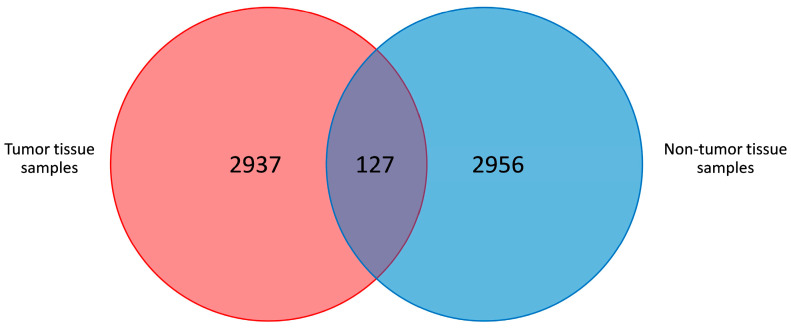
Venn diagram of the shared OTUs among tumor and non-tumor adjacent tissue samples.

**Figure 2 cancers-16-04008-f002:**
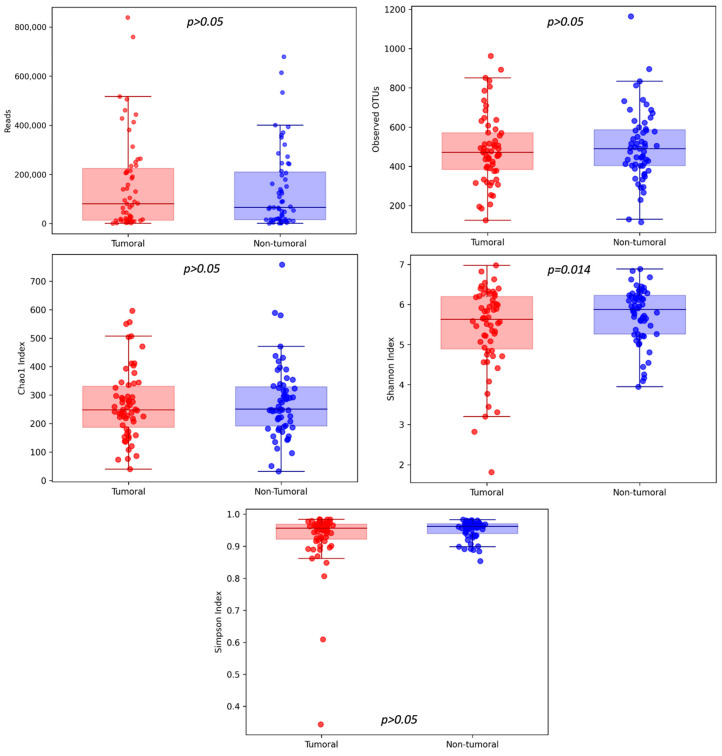
Community richness (number of OTUs, Chao1 index) and diversity and evenness (Shannon and Simpson indexes) were analyzed between tumoral and non-tumoral groups, with statistically significant differences discovered in the case of the Shannon Index.

**Figure 3 cancers-16-04008-f003:**
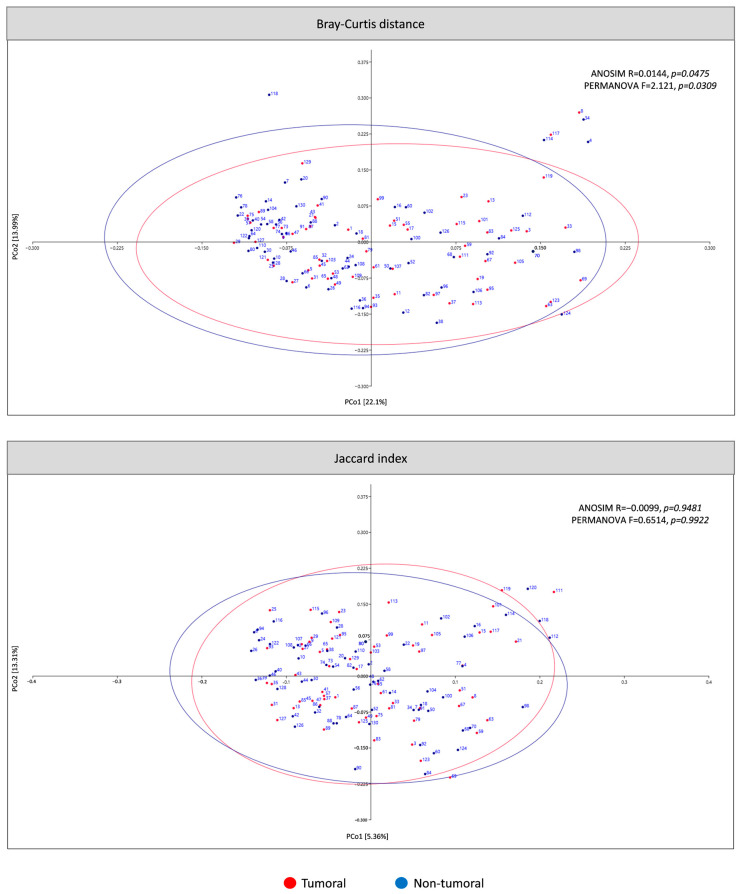
β-diversity of microbiota in tumoral compared to non-tumoral groups of CRC patients. The principal coordinates analysis (PCoA) plots were based on two metrics: Bray–Curtis dissimilarity and Jaccard index. Ellipses represent the area in which the sample is expected to be with a 95% confidence level.

**Figure 4 cancers-16-04008-f004:**
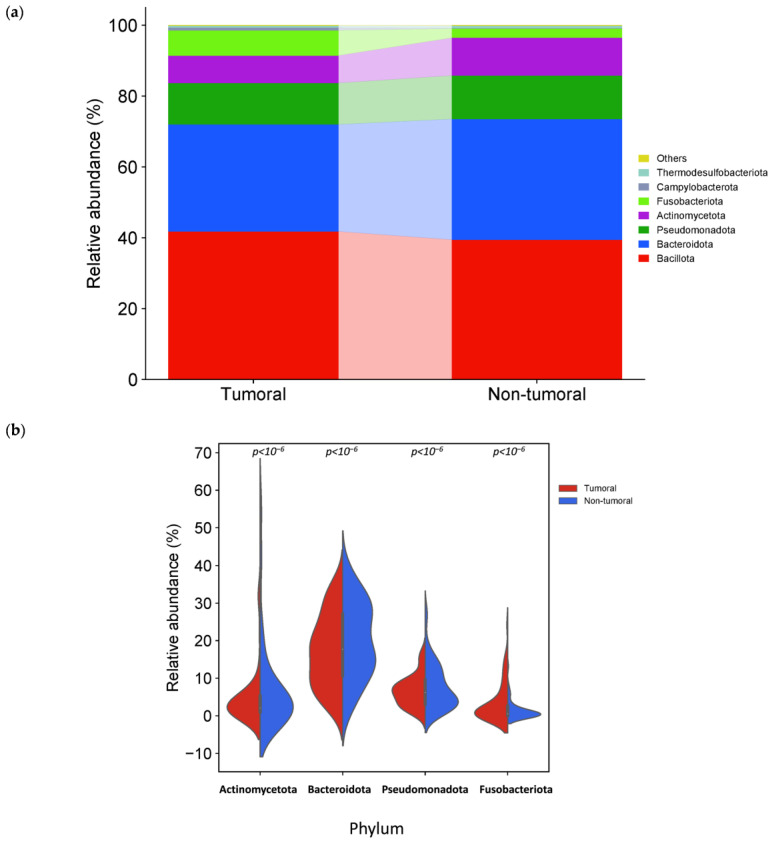
Different bacterial distribution among tumor and non-tumor adjacent tissue samples at the phylum level. (**a**) Stacked bar plots of bacterial taxa distribution at the phylum level. (**b**) Violin plot representing the relative abundance of the phylum significantly different between tumor and non-tumor tissues: Actinomycetota, Bacteroidota, Pseudomonadota and Fusobacteriota.

**Figure 5 cancers-16-04008-f005:**
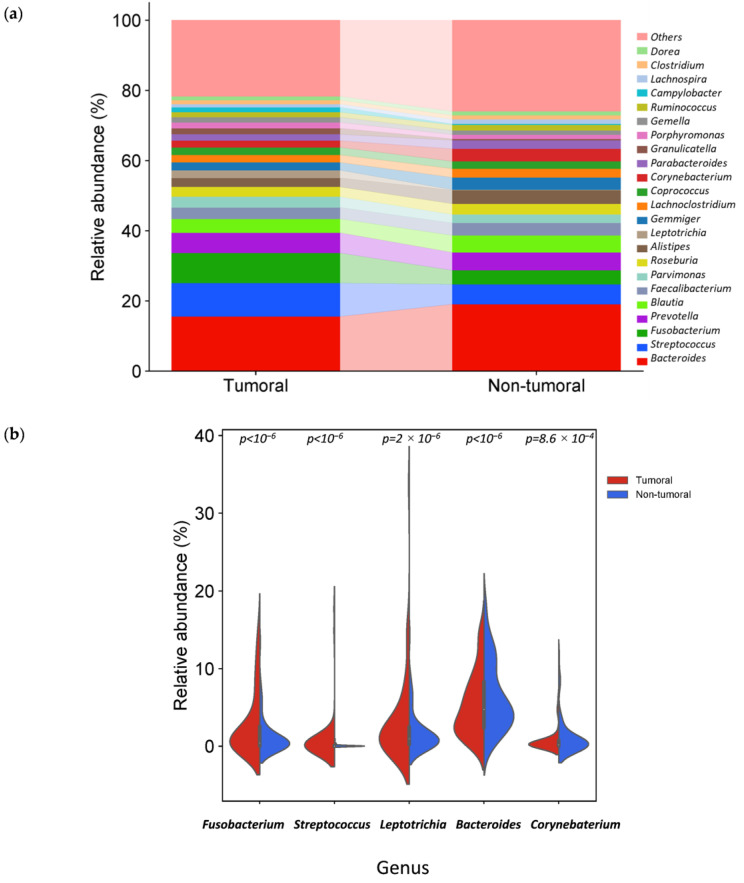
Different bacterial distribution among tumor and non-tumor adjacent tissue samples at the genus level. (**a**) Stacked bar plots of bacterial taxa distribution at the genus level. (**b**) Violin plot representing the relative abundance of the genus significantly different between tumor and non-tumor tissues: *Fusobacterium*, *Leptotrichia*, *Streptococcus*, *Bacteroidetes* and *Corynebacterium*.

**Figure 6 cancers-16-04008-f006:**
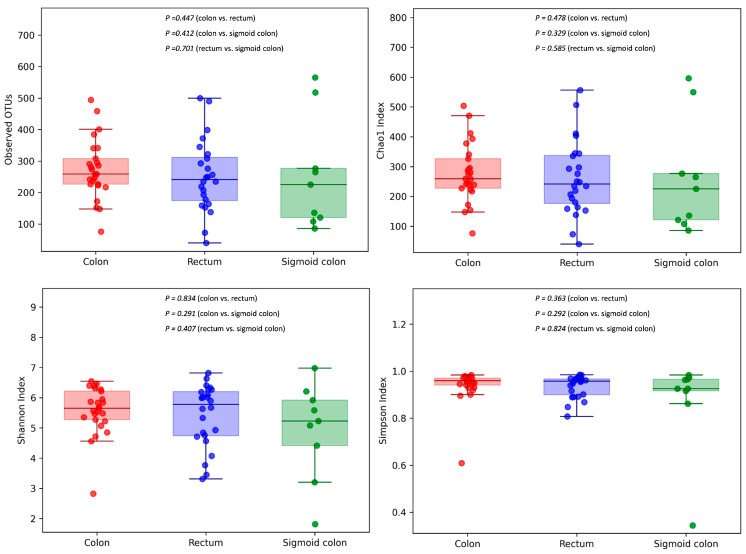
Community richness (number of OTUs, Chao1 index) and diversity and evenness (Shannon and Simpson indexes) were analyzed between CRC tumor tissue samples from the colon, rectum and sigmoid colon.

**Table 1 cancers-16-04008-t001:** Over-represented bacterial taxa in tumor compared to non-tumor adjacent tissue samples.

Taxa Name	Rank	Mean of Tumor Relative Abundance (%)	Mean of Non-Tumor Relative Abundance (%)	*p*-Value	Log 2 Fold Change	Occurrence in Tumor (%)	Occurrence in Non-Tumor (%)
Fusobacteriota	Phylum	3.49	1.38	<10^−6^	1.33	100	100
Bacilli	Class	4.09	2.47	<10^−6^	0.72	100	100
Fusobacteriia	Class	3.48	1.39	<10^−6^	1.33	100	100
Fusobacteriales	Order	3.49	1.39	<10^−6^	1.33	100	100
Lactobacillales	Order	4.25	2.44	<10^−6^	0.80	100	100
Fusobacteriaceae	Family	2.18	1.01	<10^−6^	1.11	100	100
Leptotrichiaceae	Family	0.91	0.08	<10^−6^	3.58	66.1	57.6
Streptococcaceae	Family	3.17	1.89	<10^−6^	0.75	100	100
*Fusobacterium*	Genus	2.89	1.34	<10^−6^	1.10	100	100
*Leptotrichia*	Genus	0.75	0.06	2 × 10^−6^	3.59	66.1	57.6
*Streptococcus*	Genus	3.23	1.90	<10^−6^	0.77	100	100
*Fusobacterium nucleatum*	Species	2.34	1.03	<10^−6^	1.19	100	100
*Fusobacterium polymorphum*	Species	0.62	0.13	5.3 × 10^−3^	2.29	76.3	72.9
*Streptococcus periodonticum*	Species	2.25	0.99	<10^−6^	1.19	100	100

Taxonomic name refers to the current name in the NCBI Taxonomy Browser (https://www.ncbi.nlm.nih.gov/Taxonomy/Browser/wwwtax.cgi; accessed on 19 September 2024).

**Table 2 cancers-16-04008-t002:** Under-represented bacterial taxa in tumor compared to non-tumor adjacent tissue samples.

Taxa Name	Rank	Mean of Tumor Relative Abundance (%)	Mean of Non-Tumor Relative Abundance (%)	*p*-Value	Log 2 Fold Change	Occurrence in Tumor (%)	Occurrence in Non-Tumor (%)
Actinomycetota	Phylum	4.76	7.39	<10^−6^	−0.64	100	100
Bacteroidota	Phylum	17.12	20.08	<10^−6^	−0.23	100	100
Pseudomonadota	Phylum	6.60	7.71	<10^−6^	−0.22	100	100
Actinomycetes	Class	2.72	4.39	<10^−6^	−0.69	100	98.3
Alphaproteobacteria	Class	0.50	1.31	<10^−6^	−1.40	81.4	88.1
Bacteroidia	Class	11.42	13.35	<10^−6^	−0.23	100	100
Betaproteobacteria	Class	1.72	2.66	<10^−6^	−0.62	98.3	98.3
Clostridia	Class	18.19	20.25	<10^−6^	−0.15	100	100
Bacteroidales	Order	11.75	13.74	<10^−6^	−0.23	100	100
Eubacteriales	Order	14.16	15.77	<10^−6^	−0.15	100	100
Mycobacteriales	Order	1.08	2.03	<10^−6^	−0.92	96.6	98.3
Bacteroidaceae	Family	4.36	5.32	<10^−6^	−0.29	100	100
Corynebacteriaceae	Family	0.64	1.19	8.6 × 10^−4^	−0.89	96.6	96.6
Lachnospiraceae	Family	7.48	8.46	<10^−6^	−0.18	100	100
Propionibacteriaceae	Family	1.20	1.85	1 × 10^−5^	−0.62	88.1	93.2
*Bacteroides*	Genus	5.25	6.41	<10^−6^	−0.29	100	100
*Corynebacterium*	Genus	0.64	1.19	8.6 × 10^−4^	−0.89	96.6	96.6

Taxonomic name refers to the current name in the NCBI Taxonomy Browser (https://www.ncbi.nlm.nih.gov/Taxonomy/Browser/wwwtax.cgi; accessed on 19 September 2024).

**Table 3 cancers-16-04008-t003:** ANOVA and PERMANOVA test for β-diversity of microbiota between tumor tissue samples with different locations.

	β-Diversity					
	Colon vs. Rectum		Colon vs. Sigmoid Colon		Rectum vs. Sigmoid Colon	
Metric	ANOSIM	PERMANOVA	ANOSIM	PERMANOVA	ANOSIM	PERMANOVA
Bray–Curtis	R = 0.02825 *p* = 0.0978	F = 1.821 *p* = 0.0624	R = 0.1895 *p* = 0.054	F = 1.087 *p* = 0.335	R = 0.099 *p* = 0.1587	F = 0.7444 *p* = 0.6527
Jaccard	R = 0.05181 *p* = 0.0285	F = 1.34 *p* = 0.0649	R = 0.3916 *p* = 0.0016	F = 1.422 *p* = 0.0534	R = 0.2703 *p* = 0.0095	F = 1.131 *p* = 0.2193

## Data Availability

The original contributions presented in the study are included in the article/Appendix A; further inquiries can be directed to the corresponding author.

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
