# Peer review of "New Insights into Mucosa-Associated Microbiota in Paired Tumor and Non-Tumor Adjacent Mucosal Tissues in Colorectal Cancer Patients"

_cancers, 2024, doi:10.3390/cancers16234008_

Round 1

Reviewer 1 Report

Comments and Suggestions for Authors

This article detected the microbiota composition in paired tumor and non-tumor adjacent tissue samples from 65 colorectal cancer patients, and revealed differences in microbial community composition between different tumor locations, providing new insights into understanding the microbiota changes associated with colorectal cancer. However, the differences in microbial communities between colorectal cancer tumor tissues and non-tumor tissues have been recognized. While I commend the author for the effort invested in this manuscript, unfortunately, I believe that the manuscript lacks a certain degree of novelty in its conception. In addition, many contents in this article have not been clearly described, and there are also deficiencies in language expression and format. Therefore, I believe that the author should make extensive revisions to this manuscript.

1. Although the author has provided a detailed description in the abstract section, I still suggest that the author revise it into a structured abstract that succinctly summarizes the research objectives, methods, results, and conclusions, so that readers can quickly understand the core content of the article.

2. The introduction section of the paper provides a relatively brief explanation of the research background and significance, which does not adequately demonstrate the urgency and importance of the study. It is recommended that the authors provide a review of existing research on the relationship between colorectal cancer and microbial communities, as well as the specific contributions of this study in filling knowledge gaps or improving existing methods.

3. Microbiota in colorectal cancer tumor tissues are closely related to genetic factors, dietary habits, disease status, antibiotic use, and other factors. However, the manuscript does not specify the inclusion and exclusion criteria for the research subjects. It is suggested that the author should provide a detailed explanation in the methods section to ensure the reliability of the results in the article.

4. In line 113, the author describes "27 cases in the colon, 27 cases in the rectum, 9 cases in the sigmoid colon, and 1 case in the cecum" . The total number of cases here is 64, which does not correspond to the 65 patients mentioned in the article.

5. Please provide a detailed explanation of the sample collection method and explain the source of non-tumor tissue samples.

6. The article title refers to the mucosa-associated microbiota, but the article focuses on detecting the microbiota in both tumor and non-tumor tissues. There seems to be some confusion between these two concepts, and it is recommended that the author clarify them accurately.

7. The author only compared colorectal cancer tumor tissue with non-tumor tissue, and failed to analyze the tumor microbiota in conjunction with clinical indicators and other influencing factors. The data lacks representativeness and repeatability.

8. Please provide clearer and more complete images.

9. Despite listing microbial community differences between tumor and non-tumor tissues, the results section lacks depth in interpreting their biological significance. The author should discuss these changes more thoroughly, exploring their carcinogenic or protective mechanisms.

10. Although the manuscript has a conclusion section, the summary is not concise and comprehensive enough. It is recommended that the author provide a succinct and clear summary of the main findings in the conclusion section, emphasizing its scientific significance and practical value.

11. There are some language usages in the manuscript that are not accurate or clear enough. It is recommended that the author carefully proofread the manuscript to ensure the accuracy and clarity of language usage.

12. There are inconsistencies in the citation format of the manuscript. It is recommended that the author uniformly format the citations in accordance with the journal's citation standards.

Author Response

Manuscript ID: cancers-3298359 (Response to Reviewer 1)

Comment: This article detected the microbiota composition in paired tumor and non-tumor adjacent tissue samples from 65 colorectal cancer patients, and revealed differences in microbial community composition between different tumor locations, providing new insights into understanding the microbiota changes associated with colorectal cancer. However, the differences in microbial communities between colorectal cancer tumor tissues and non-tumor tissues have been recognized. While I commend the author for the effort invested in this manuscript, unfortunately, I believe that the manuscript lacks a certain degree of novelty in its conception. In addition, many contents in this article have not been clearly described, and there are also deficiencies in language expression and format. Therefore, I believe that the author should make extensive revisions to this manuscript.

Thank you. We consider that your suggestions are very appropriate and significantly improve the work. We include the answers after each comment.

1.Although the author has provided a detailed description in the abstract section, I still suggest that the author revise it into a structured abstract that succinctly summarizes the research objectives, methods, results, and conclusions, so that readers can quickly understand the core content of the article.

We have changed the abstract into a more structured one, clearly indicating the background, objective, methods, results, and conclusions sections [page number 1, paragraph 2, and lines 26-46].

We have improved the writing of specific sentences:

  • [page number 1, paragraph 2, and lines 29-35]: This article aims to address these limitations, providing new insights into the microbiota associated with CRC pathogenesis and implementing its analyses in personalized medicine. Methods: To that aim, we evaluate differences in the bacterial composition of 130 paired tumor and non-tumor adjacent tissues from a cohort of CRC patients from the Biobank of the University of Navarra, Spain. The V3-V4 region of the 16S rRNA gene was amplified, sequenced using the MinION platform, and taxonomically classified using the NCBI database.
  • [page number 1, paragraph 2, and lines 45-46]: Overall, these results contribute to a better understanding of the CRC-associated microbiota, addressing critical barriers to its implementation in personalized medicine.

2.The introduction section of the paper provides a relatively brief explanation of the research background and significance, which does not adequately demonstrate the urgency and importance of the study. It is recommended that the authors provide a review of existing research on the relationship between colorectal cancer and microbial communities, as well as the specific contributions of this study in filling knowledge gaps or improving existing methods.

We have included the following information related to this comment:

  • [page number 3, paragraph 3, and lines 66-73]: These pathogens may contribute to colorectal carcinogenesis directly by damaging DNA and stimulating colonocyte proliferation or indirectly by promoting a favorable environment for CRC development [15,16]. Otherwise, some bacteria in the gut microbiota exert protective effects against CRC. The main mechanisms by which the gut microbiota plays a key role in colorectal carcinogenesis include intestinal permeability regulation, inflammation and immune response modulation, biofilm formation, genotoxin production, virulence factors, oxidative stress and metabolite production [17].
  • [page number 3, paragraph 4, and lines 74-80]: There are two hypotheses regarding the role of the gut microbiota in colorectal carcinogenesis. The “alpha-bug” hypothesis suggests that certain pro-oncogenic microorganisms can displace cancer-protective bacteria and colonize the tumor persistently, creating an environment favorable for tumorigenesis [18]. On the other hand, the driver-passenger model suggests that “driver” bacteria that initiate CRC are then replaced by “passenger” bacteria with growth advantages in the tumor microenvironment (TME) that may exert promoting or inhibiting effects on the tumor progression [19].
  1. Microbiota in colorectal cancer tumor tissues are closely related to genetic factors, dietary habits, disease status, antibiotic use, and other factors. However, the manuscript does not specify the inclusion and exclusion criteria for the research subjects. It is suggested that the author should provide a detailed explanation in the methods section to ensure the reliability of the results in the article.

The information has been included in the methods section [page number 4, paragraph 6, and lines 132-134]: The general information (age and gender) and clinical data (tumor location, tumor differentiation and tumor stage) of samples are shown in Supplementary Table S1.

As this information has been included as Supplementary Table S1, the order of the supplementary tables has been updated throughout the manuscript [page number 13, paragraphs 1-3, and lines 365-381], [page number 18, Supplementary Materials, and lines 650-655]. The numbers should also be updated in the supplementary material tables.

We have also included the following sentence in the Acknowledgments section (lines 668-669): We particularly acknowledge the patients for their participation and the Biobank of the University of Navarra for its collaboration.

  1. In line 113, the author describes "27 cases in the colon, 27 cases in the rectum, 9 cases in the sigmoid colon, and 1 case in the cecum" . The total number of cases here is 64, which does not correspond to the 65 patients mentioned in the article.

You are right. It has been corrected [page number 4, paragraph 6, and line 135]: […] the colon for 27, the rectum for 28, the sigmoid colon for 9 and the cecum for 1.

  1. Please provide a detailed explanation of the sample collection method and explain the source of non-tumor tissue samples.

We have included this explanatory paragraph [page number 4, paragraph 5, and lines 127-131]: The pathologist selected, if possible, a fragment of tumor tissue and a fragment of non-tumor adjacent mucosal tissues. A Biobank technician, working in sterile conditions and with the material on dry ice, cut the selected tissue into small 2-3mm square fragments placed in a cryotube for immediate freezing in dry ice. All were registered and stored at -80°C at the Biobank until DNA extraction.

  1. The article title refers to the mucosa-associated microbiota, but the article focuses on detecting the microbiota in both tumor and non-tumor tissues. There seems to be some confusion between these two concepts, and it is recommended that the author clarify them accurately.

You are right. For better clarity and coherence, we have included this data in the Title: New Insights into Mucosa-Associated Microbiota in Paired Tumor and Non-Tumor Adjacent Mucosal Tissues in Colorectal Cancer Patients.

We have also indicated it in the Materials and Methods section [page number 4, paragraph 5, and lines 127-128]: The pathologist selected, if possible, a fragment of tumor tissue and a fragment of non-tumor adjacent mucosal tissues.

  1. The author only compared colorectal cancer tumor tissue with non-tumor tissue, and failed to analyze the tumor microbiota in conjunction with clinical indicators and other influencing factors. The data lacks representativeness and repeatability.

We have included the following information in the Material and Methods section [page number 4, paragraph 6, and lines 132-134]: The general information (age and gender) and clinical data (tumor location, tumor differentiation and tumor stage) of samples are shown in Supplementary Table S1.

  1. Please provide clearer and more complete images. We have improved the images. If this mistake continues, do not hesitate to tell us. Thank you.
  2. Despite listing microbial community differences between tumor and non-tumor tissues, the results section lacks depth in interpreting their biological significance. The author should discuss these changes more thoroughly, exploring their carcinogenic or protective mechanisms.
  • [page number 14, paragraph 4, and lines 441-442]: These procarcinogenic effects could be mediated by components such as lipopolysaccharide and adhesins like FadA and Fap2
  • [page number 15, paragraph 3, and lines 480-484]: Interestingly, a bacterium of the genus Streptococcus, gallolyticus, is a proinflammatory species remarkably associated with CRC. This bacterium can express collagen-binding proteins such as Pil 1, allowing it to colonize tissues and induce the secretion of proinflammatory mediators that can promote CRC [11].
  • [page number 15, paragraph 6, and lines 503-506]: Differences in the microbial profiles observed between tumor and non-tumor tissues could reflect the role of certain microbiota members in the initiation and development of CRC or changes associated with the TME. Two main hypotheses have been proposed to explain these interactions.
  • [page number 15, paragraph 7, and lines 507-512]: The “alpha-bug” hypothesis proposed by Sears and Pardoll suggests that detrimental members of the microbiota, such as enterotoxigenic Bacteroides fragilis (ETBF), gallolyticus, superoxide-producing Enterococcus faecalis and Escherichia coli, may act as cancer initiators by directly inducing alterations in colonic epithelial cells and remodelling the colonic microbial community to promote these alterations and disrupt immune responses [18].
  • [page number 16, paragraph 1, and lines 513-520]: In contrast, the bacterial driver-passenger model proposed by Tjalsma et al. suggests that certain commensal bacteria (bacterial drivers), such as Enterococcus faecalis, can cause epithelial DNA damage, leading to CRC initiation. Next, the process of tumorigenesis induces modifications in the microenvironment that benefit relatively poor colonizing bacteria (bacterial passengers) [19]. Bacterial passengers have been proposed to include opportunistic pathogens that feed in the tumour (Fusobacterium or Streptococcus), intestinal commensal or probiotic bacteria (Coriobacteriaceae family), and bacteria that have a competitive advantage in the TME [19].
  1. Although the manuscript has a conclusion section, the summary is not concise and comprehensive enough. It is recommended that the author provide a succinct and clear summary of the main findings in the conclusion section, emphasizing its scientific significance and practical value.

A succinct and clearer summary of the main findings has been included in the conclusion section. The scientific significance and practical value of the results has been highlighted.

  • [page number 18, paragraph 3, and lines 638-645]: It comprises the analysis of 130 paired tumor and non-tumor adjacent tissue samples in different locations by larger amplicon sizes, including the V3-V4 region of 16S rRNA gene analyzed by MinION suitable sequence platform, and using the NCBI database, that increases the amplitude and resolution in sequence taxonomy classification. These methodological advances agree earlier findings of microbiota composition differences between tumor and non-tumor tissues. They also reveal the possible involvement of specific taxa such as periodonticus and Corynebacterium, in CRC biology for the first time.
  • [page number 18, paragraph 4, and lines 646-648]: Therefore, the results obtained in this study facilitate the implementation of individual gut microbiota analysis in personalized medicine. This approach allows the development of therapeutic strategies for CRC that consider this essential component of TME [20].

  1. There are some language usages in the manuscript that are not accurate or clear enough. It is recommended that the author carefully proofread the manuscript to ensure the accuracy and clarity of language usage.

Thank you so much. Language usage has been revised, and sentences have been rewritten for better clarity. The changes made are indicated below:

  • [page number 3, paragraph 8, and line 96]: “TME” instead of “tumor microenvironment” (already mentioned before).
  • [page number 13, paragraph 5, and line 397]:“favors” instead of “favours”.
  • [page number 14, paragraph 4, and lines 431, 432, 435, 436, 437, 442 ]:“" nucleatum" instead of "F. nucleatum".
  • [page number 15, paragraph 4, and lines 488-489]:“: “Although a Pseudomonadota increase […]” instead of “Although an increase in Pseudomonadota is […]”.
  • [page number 17, paragraph 4, and line 588-589]:“: “It would be advisable to analyze more samples from each patient for future research […]” instead of “For future research, it would be advisable to analyze more samples from each patient to […]”
  1. There are inconsistencies in the citation format of the manuscript. It is recommended that the author uniformly format the citations in accordance with the journal's citation standards.

You are right. We have revised it in every section for all the citations and unified the format as the American Chemical Society recommended using Multidisciplinary Digital Publishing Institute Zotero Citation Style for MDPI journals. Specifically:

  • The website reference previously cited in the Introduction as "(IARC 2023, https://gco.iarc.fr/)" has been updated to "[1] [page number 3, paragraph 1, and line 56] and added to the reference list with the required format.

Thank you. Kind regards!

Reviewer 2 Report

Comments and Suggestions for Authors

It's a very interesting research article about Mucosa-Associated Microbiota in Tumor Tissues in Colorectal Cancer Patients.

1. The introduction can be improved by describing more about co-relation between microbiota and tumor progression.

2. The inclusion and exclusion criteria for selecting patients should be given.  What was the stage of the cancer patients, age and sex, therapy going on or not. All these demographic information seems to be missing that can be included as supplementary information.

Author Response

Manuscript ID: cancers-3298359 (Response to Reviewer 2)

Comment: It's a very interesting research article about Mucosa-Associated Microbiota in Tumor Tissues in Colorectal Cancer Patients.

Thank you for your words and your suggestions. We have included your comments to substantially improve the work. We include the answers after each comment.

  1. The introduction can be improved by describing the co-relation between microbiota and tumor progression.

We have included this information about co-relation between microbiota and tumor progression:

  • [page number 3, paragraph 3, and lines 66-73]: These pathogens may contribute to colorectal carcinogenesis directly by damaging DNA and stimulating colonocyte proliferation or indirectly by promoting a favorable environment for CRC development [15,16]. Otherwise, some bacteria in the gut microbiota exert protective effects against CRC. The main mechanisms by which the gut microbiota plays a key role in colorectal carcinogenesis include intestinal permeability regulation, inflammation and immune response modulation, biofilm formation, genotoxin production, virulence factors, oxidative stress and metabolite production [17].
  • [page number 3, paragraph 4, and lines 74-80]: There are two hypotheses regarding the role of the gut microbiota in colorectal carcinogenesis. The “alpha-bug” hypothesis suggests that certain pro-oncogenic microorganisms can displace cancer-protective bacteria and colonize the tumor persistently, creating an environment favorable for tumorigenesis [18]. On the other hand, the driver-passenger model suggests that “driver” bacteria that initiate CRC are then replaced by “passenger” bacteria with growth advantages in the tumor microenvironment (TME) that may exert promoting or inhibiting effects on the tumor progression [19].
  1. The inclusion and exclusion criteria for selecting patients should be given.  What was the stage of the cancer patients, age and sex, therapy going on or not. All these demographic information seems to be missing that can be included as supplementary information.

The information has been included in the methods section [page number 4, paragraph 6, and lines 132-134]: The general information (age and gender) and clinical data (tumor location, tumor differentiation and tumor stage) of samples are shown in Supplementary Table S1.

As this information has been included as Supplementary Table S1, the order of the supplementary tables has been updated throughout the manuscript [page number 13, paragraphs 1-3, and lines 365-381], [page number 18, Supplementary Materials, and lines 650-655]. The numbers should also be updated in the supplementary material tables.

We have also included the following sentence in the Acknowledgments section (lines 668-669): We particularly acknowledge the patients for their participation and the Biobank of the University of Navarra for its collaboration.

Thank you. Kind regards!

Reviewer 3 Report

Comments and Suggestions for Authors

This study is important in advancing our knowledge of CRC-associated microbiota and the potential implications for treatment. The methods adopted such as MinION sequencing and NCBI taxonomic classification add robustness to their results. While the authors successfully identify distinct microbial profiles in tumor versus non-tumor tissues, it is worth considering that these profiles may, in part, reflect the local tumor microenvironment rather than being primary drivers of tumor development. The authors should have considered this possibiligy and discussed it.

-----------------------------------------------------------------

The research explores whether specific bacterial taxa are enriched or diminished in tumor tissues, with the potential implication of CRC occurrence. This focus is highly original and relevant to CRC research, particularly as most microbiota studies in this field have relied on fecal samples.  The authors tackle the specific role of the mucosa-associated microbiota in CRC by analyzing paired tumor and non-tumor tissues, addressing a gap in the current understanding of how local microbiota may be linked to cancer. The use of the MinION platform alongside the NCBI taxonomy database also enhances the taxonomic resolution, allowing the authors to report previously unrecognized microbial associations at the species and strain levels.  However, including a control group of tissue samples from CRC-free individuals would enhance the findings. Such a control group would help determine if the observed microbial differences are specific to CRC or simply reflect normal tissue variability. This consideration is noted by the authors in their Discussion section. Additionally, incorporating longitudinal sampling to track microbiota changes over time in the same patients would help clarify whether specific bacteria play a role in tumor progression or if they are simply responding to the tumor environment. The conclusions presented are consistent with the evidence and align with the main research question, successfully identifying specific microbial profiles in tumor tissues. However, while microbial profiles differ between tumor and non-tumor tissues, these differences may reflect the distinct tumor microenvironment rather than directly contributing to tumor initiation or progression. Additional references could strengthen the discussion, especially studies examining the impact of the tumor microenvironment on microbiota composition. 

Author Response

Manuscript ID: cancers-3298359 (Response to Reviewer 3)

Comment: This study is important in advancing our knowledge of CRC-associated microbiota and the potential implications for treatment. The methods adopted such as MinION sequencing and NCBI taxonomic classification add robustness to their results. While the authors successfully identify distinct microbial profiles in tumor versus non-tumor tissues, it is worth considering that these profiles may, in part, reflect the local tumor microenvironment rather than being primary drivers of tumor development. The authors should have considered this possibiligy and discussed it.

The research explores whether specific bacterial taxa are enriched or diminished in tumor tissues, with the potential implication of CRC occurrence. This focus is highly original and relevant to CRC research, particularly as most microbiota studies in this field have relied on fecal samples.  The authors tackle the specific role of the mucosa-associated microbiota in CRC by analyzing paired tumor and non-tumor tissues, addressing a gap in the current understanding of how local microbiota may be linked to cancer. The use of the MinION platform alongside the NCBI taxonomy database also enhances the taxonomic resolution, allowing the authors to report previously unrecognized microbial associations at the species and strain levels.  However, including a control group of tissue samples from CRC-free individuals would enhance the findings. Such a control group would help determine if the observed microbial differences are specific to CRC or simply reflect normal tissue variability. This consideration is noted by the authors in their Discussion section. Additionally, incorporating longitudinal sampling to track microbiota changes over time in the same patients would help clarify whether specific bacteria play a role in tumor progression or if they are simply responding to the tumor environment. The conclusions presented are consistent with the evidence and align with the main research question, successfully identifying specific microbial profiles in tumor tissues. However, while microbial profiles differ between tumor and non-tumor tissues, these differences may reflect the distinct tumor microenvironment rather than directly contributing to tumor initiation or progression. Additional references could strengthen the discussion, especially studies examining the impact of the tumor microenvironment on microbiota composition. 

Thank you for your contribution and revision of the manuscript. Your suggestions are important and bring an improvement to the work's content. We have included this information and references:

  • [page number 15, paragraph 6, and lines 503-506]: Differences in the microbial profiles observed between tumor and non-tumor tissues could reflect the role of certain microbiota members in the initiation and development of CRC or changes associated with the TME. Two main hypotheses have been proposed to explain these interactions.
  • [page number 15, paragraph 7, and lines 507-512]: The “alpha-bug” hypothesis proposed by Sears and Pardoll suggests that detrimental members of the microbiota, such as enterotoxigenic Bacteroides fragilis (ETBF), gallolyticus, superoxide-producing Enterococcus faecalis and Escherichia coli, may act as cancer initiators by directly inducing alterations in colonic epithelial cells and remodelling the colonic microbial community to promote these alterations and disrupt immune responses [18].
  • [page number 16, paragraph 1, and lines 513-520]: In contrast, the bacterial driver-passenger model proposed by Tjalsma et al. suggests that certain commensal bacteria (bacterial drivers), such as Enterococcus faecalis, can cause epithelial DNA damage, leading to CRC initiation. Next, the process of tumorigenesis induces modifications in the microenvironment that benefit relatively poor colonizing bacteria (bacterial passengers) [19]. Bacterial passengers have been proposed to include opportunistic pathogens that feed in the tumour (Fusobacterium or Streptococcus), intestinal commensal or probiotic bacteria (Coriobacteriaceae family), and bacteria that have a competitive advantage in the TME [19].
  • [page number 17, paragraph 4, and lines 591-594]: In addition, future longitudinal studies looking at changes in the microbiota over time would provide a better understanding of whether specific bacteria play a role in the initiation and progression of CRC or whether they are responding to TME. Finally, considering the pros and cons of analyzing faecal or tissue samples, combining both seems valuable for future research.

We have also briefly mentioned this point in the Introduction:

  • [page number 3, paragraph 4, and lines 74-80]: There are two hypotheses regarding the role of the gut microbiota in colorectal carcinogenesis. The “alpha-bug” hypothesis suggests that certain pro-oncogenic microorganisms can displace cancer-protective bacteria and colonize the tumor persistently, creating an environment favorable for tumorigenesis [18]. Otherwise, the driver-passenger model suggests that “driver” bacteria that initiate CRC are then replaced by “passenger” bacteria with growth advantages in the tumor microenvironment (TME) that may exert promoting or inhibiting effects on the tumor progression [19].

Thank you. Kind regards!

Round 2

Reviewer 1 Report

Comments and Suggestions for Authors

the paper can be accepted